# Experimental Evidence for the Anti-Metastatic Action of Ginsenoside Rg3: A Systematic Review

**DOI:** 10.3390/ijms23169077

**Published:** 2022-08-13

**Authors:** Hyeon-Muk Oh, Chong-Kwan Cho, Chang-Gue Son

**Affiliations:** 1College of Korean Medicine, Daejeon University, Daejeon 35235, Korea; 2East-West Cancer Center, Daejeon Korean Medicine Hospital of Daejeon University, Daejeon 35235, Korea; 3Liver and Immunology Research Center, Daejeon Korean Medicine Hospital of Daejeon University, Daejeon 35235, Korea

**Keywords:** ginsenoside Rg3, cancer metastasis, systematic review

## Abstract

Cancer metastasis is the leading cause of death in cancer patients. Due to the limitations of conventional cancer treatment, such as chemotherapy, there is a need for novel therapeutics to prevent metastasis. Ginsenoside Rg3, a major active component of *Panax ginseng* C.A. Meyer, inhibits tumor growth and has the potential to prevent tumor metastasis. Herein, we systematically reviewed the anti-metastatic effects of Rg3 from experimental studies. We searched for articles in three research databases, MEDLINE (PubMed), EMBASE, and the Cochrane Central Register of Controlled Trials (CENTRAL) through March 2022. In total, 14 studies (eight animal and six in vitro) provide data on the anti-metastatic effects of Rg3 and the relevant mechanisms. The major anti-metastatic mechanisms of Rg3 involve cancer stemness, epithelial mesenchymal transition (EMT) behavior, and angiogenesis. Taken together, Rg3 would be one of the herbal resources in anti-metastatic drug developments through further well-designed investigations and clinical studies. Our review provides valuable reference data for Rg3-derived studies targeting tumor metastasis.

## 1. Introduction

Cancer metastasis is the main cause of cancer-related deaths, as 66% of the 10.0 million cancer deaths worldwide in 2020 were due to metastasis [1,2]. Despite scientific and technological advances, there has been no significant decrease in the mortality rate among metastatic cancer patients over the past decade [3]. The 5-year survival rate of metastatic cancer was significantly lower than primary cancer; nearly 0.25-fold for breast cancer, 0.13-fold for colorectal cancer, and 0.08-fold for pancreatic cancer [4]. Accordingly, reducing the risk of cancer metastasis is a crucial factor in the survival of cancer patients and cancer treatment [5].

In cancer treatment, the early detection and complete removal of a tumor is optimal [6,7]. Its failure, however, is very common, and approximately 40% of cancer patients are known to progress into metastasis during their treatment process after diagnosis [8]. Cancer metastasis is a complex and multistep process that consists of dissemination, intravasation, circulation, extravasation, and colonization [9]. As the tumor progresses into an advanced stage, the genetic alterations leading to pro-metastatic characteristics are obtained, which trigger the epithelial mesenchymal transition (EMT) known to induce tumor metastasis [10,11]. In addition, tumor microenvironments (TMEs) such as the extracellular matrix structure, matrix metalloproteinases (MMPs), growth factors, and chemokines are also known to play important roles in cancer metastasis [12].

Accordingly, there have been strong efforts to develop appropriate anti-metastatic drugs [13]. To date, two major anti-metastatic drugs, an anti-angiogenesis drug and an MMPs inhibitor, have been approved by the Food and Drug Administration in the USA [14]. They target MMPs, which play an important role in the processes of intravasation and angiogenesis, required for colonization and tumor growth in secondary metastatic organs [15,16]. Although the clinical trials did not show enough efficacy on anti-metastasis [17], the development of anti-metastatic drugs targeting these two mechanisms is in progress [18].

On the other hand, there is growing evidence that several natural compounds such as the ginsenoside in *Panax ginseng* C.A. Meyer (*P. ginseng*) and curcumin in *Curcuma longa* L. have the potential to prevent cancer metastasis [19,20,21]. *P. ginseng* has traditionally been used to treat various diseases for over a thousand years in Asian countries [22]. Among the numerous active compounds of *P. ginseng*, ginsenoside Rg3 has been most prevalently studied for its anticancer effects in many types of cancers, including breast cancer, gastrointestinal cancer, and prostate cancer [23,24,25]. Moreover, Rg3 has been known to enhance host immunity, reduce the adverse effects of chemotherapies, and increase survival periods, according to clinical studies [26,27]. Some groups also reported the modulations of TMEs, including anti-angiogenesis from animal studies [28,29]. In addition, some studies showed the inactivation of cancer stem cell (CSC) and EMT by Rg3, which would anticipate the anti-metastatic potentials of Rg3 [30,31].

In order to provide the basis for antimetastatic drug development using ginsenosides, we herein systematically reviewed the anti-metastatic effects of Rg3, a representative active compound of *P. ginseng*.

## 2. Materials and Methods

### 2.1. Search Strategy and Selection Criteria

Three electronic databases were searched for the systematic literature survey, including MEDLINE (PubMed), EMBASE, and the Cochrane Central Register of Controlled Trials (CENTRAL), limited to papers published on or before March 2022. MeSH and Emtree searching were applied too. The search was conducted by combining keywords related to ginsenoside Rg3 and metastasis and its combination.

The inclusion criteria for studies were the evaluation of the anti-metastatic effects of Rg3 in an experimental study either in vivo or in vitro. We excluded studies only focusing on the inhibition of cancer initiation/progression by Rg3. Articles without full text were also excluded.

### 2.2. Data Extraction and Analysis

We extracted the following details: name of first author, publication year, cancer type (cell line), animal, metastatic model, target organ, type of Rg3, concentration, administration route, duration of Rg3 treatment, main outcome of study, and mechanism of actions. The authors carefully reviewed all included studies, and they are summarized in tables and figures.

## 3. Results

### 3.1. Characteristics of the Included Studies

Of the 43 related articles, 14 studies were finally selected with four animal and in vitro combined studies, four animal studies, and six in vitro studies. (Figure 1). Four types of Rg3 (ten datasets, 20 (R)-Rg3 for six datasets, 20 (S)-Rg3 for two datasets, and one of nanoparticle-conjugated Rg3) and eight different tissue-derived cancer cells (including two mutagen-inductions) were applied for targeting three organs/tissues (lung, liver, kidney, and peritoneum), respectively. All studies were conducted in Asian countries, including China (eight studies), Korea (four studies), and Japan (two studies) (Table 1).

### 3.2. Anti-Metastatic Effects of Rg3 in Animal Studies

Of the eight animal studies, seven studies (all except one study) reported statistically significant anti-metastatic effects, identifying a decreased number of metastatic nodules or colonies in the metastatic regions; lung (two melanoma, one ovarian cancer, one colorectal cancer, one thyroid cancer, and one HCC), peritoneum (one colorectal cancer), liver (two colorectal cancer), and kidney (one colorectal cancer) (Table 2). The mean concentration of Rg3 was 27.0 ± 26.1 mg/kg/day orally (four studies) and 4.0 ± 1.0 mg/kg/day subcutaneously (four studies), and the administration period was 28 ± 28 days.

### 3.3. Anti-Metastatic Mechanisms of Rg3 in the Tumor Tissues of Animal Studies

Regarding the underlying mechanisms in animal studies, the anti-angiogenic effect of Rg3 in metastasized regions most frequently appeared (six out of eight studies). Other mechanisms involved the inhibition of cancer stem cells in one colorectal liver-metastatic model and suppression of MMPs (MMP2 and MMP9) in two lung metastatic models from SKOV-3 ovarian cancer and B16 melanoma. Rg3 also suppressed tumor growth via modulation of the ERK/AKT/mTOR signaling pathway in melanoma (Table 2).

### 3.4. Anti-Invasion and Migration Effects of Rg3 in Cancer Cell Lines

Nine of the reported studies showed the anti-invasion activity of Rg3 in 18 cell lines of nine types of tumors, originating from lung, skin, colorectum, nasopharynx, liver, and bone tumors.

Similarly, eight out of ten in vitro datasets revealed the anti-migration activity of Rg3 using 17 cell lines of eight types of tumors, originating from lung, skin, colorectum, thyroid, liver, and bone tumors (Table 2).

### 3.5. Anti-Metastatic Mechanisms of Rg3 in Cancer Cell Lines

The three major anti-metastatic mechanisms of Rg3 involve cancer stemness, EMT behavior, and MMP-related activities, as summarized in Table 2.

Briefly, Rg3 repressed the cancer stemness in both colorectal cancer (LoVo, SW620, HCT116) and liver cancer cells (HepG2, MHCC-97L) via reductions in CD24, CD44, and the epithelial cell adhesion molecule (EpCAM) and activation of ARHGAP9, a tumor suppressor gene. Rg3 also modulated EMT-related molecules, such as activations of E-cadherin and Snail and suppressions of N-cadherin, Vimentin, and zing-finger E-box-binding homeobox factor (ZEB) 1 in five different cell lines: lung, nasopharynx, colorectum, and bone tumor, respectively. For EMT-related pathways, the modulations of ERK/AKT/mTOR in B16F1 melanoma cells and Wnt/β-catenin signaling in osteosarcoma cell lines (MG63, 143B) were reported. The pharmaceutical actions of Rg3 on the inhibition of MMPs were: MMP2 in six cell lines (originated from lung, thyroid, nasopharynx, colorectum, and bone tumor), MMP9 in four cell lines (lung, thyroid, nasopharynx, and bone tumor), and MMP13 in the B16F10 melanoma cell line, respectively.

## 4. Discussion

We systematically analyzed the studies for the anti-metastatic effects of ginsenoside Rg3. Rg3 was firstly identified in 1966 as one of the major active compounds in *P. ginseng* [46]. Along with the numerous beneficial effects of *P. ginseng* for subjects with sub-healthy conditions or many diseases, its active compounds have been also of interest [47,48,49,50]. *P. ginseng* has many active compounds, including more than 40 types of ginsenosides such as Rb1, Rg1, Rg3, Re, Rd, and Rh1 [51]. Of these, Rg3 has been particularly noted for its antitumor activity in various cancers, such as breast cancer, colon cancer, lung cancer, and ovarian cancer [52,53,54].

Clinical data have shown the beneficial effects of Rg3 on cancer patients. Administration of Rg3 improved the survival rates and quality of life in patients with esophageal cancer, gastric cancer, and lung cancer [55,56,57]. Rg3 also showed an adjuvant effect with conventional cancer therapy, which increased the response rate of chemotherapy in lung cancer and sensitized the effects of radiotherapy in colorectal cancer [58,59]. Another study presented the survival gain (1.7 months) in 115 metastatic lung cancer patients by Rg3 combination therapy compared to chemotherapy alone [60]. Furthermore, Rg3 ameliorated side effects such as myelosuppression in patients undergoing chemotherapy [27,61].

From the aspect of the clinical impact of metastasis during whole processes of tumor treatment, the above data may support the anti-metastatic effects of *P. ginseng* and its major active compound, Rg3. From our systematic review, a total of 14 studies reported the anti-metastatic effects of Rg3, including eight animal results and six in vitro studies. In general, lung and liver are two of the main target organs for systemic metastasis in clinics, depending on the circulation pattern and the anatomy of vessels [62]. Our results also showed that Rg3 exerted anti-metastatic effects against mainly lung (six cell line experiments from five kinds of tumors) [32,34,36,37,38,39] and liver metastasis [35,38] (Table 2). Three of eight animal studies provided evidence on the anti-metastatic activities of Rg3, using colorectal cancer targeting liver, peritoneum, lung, and kidney [33,35,38]. Owing to its reproducibility and similarity with human pathology, the colorectal cancer-derived animal model is the most representative experimental design to study the mechanistic and molecular analyses of tumor metastasis [63].

Regarding the molecular actions of Rg3′s anti-metastatic effects, Rg3 involved the major processes of tumor metastasis, such as CSC, EMT, and angiogenesis (Figure 2). CSCs, known as key factors in the initiation of metastasis by self-renewal and differentiation into multiple phenotypes [64,65], were suppressed by Rg3 in a colorectal cancer-induced liver metastasis mouse [35]. In addition, EMT is the most well-known metastatic process accelerating tumor invasion and migration, which is regulated by various factors, including the ZEB family, Snail, and Slug [66]. Rg3 modulated EMT-related molecules, such as E-cadherin, N-cadherin, Snail, Vimentin, and the ZEB family, as reported by five in vitro studies [38,40,42,43,45]. In addition, Rg3 inhibited MMPs, including MMP2 [36,37,38,40,42,43,45], MMP7 [45], MMP9 [36,37,43,45], and MMP13 [41]. These proteinase activations are an important factor that induce EMT behaviors and also promote tumor growth and metastasis [67,68]. After the disseminated cancer cells reach the distant metastatic organ, angiogenesis is essential for the tumor survival and growth [69]. Most of our animal data (six of eight studies) showed the anti-angiogenic actions of Rg3 in various type of cancers [32,33,34,35,36,37].

Additionally, various pathophysiological changes such as inflammation and oxidative stress are associated with the promotion of tumor metastasis by inducing tumor cell proliferation, malignant phenotype, and genetic instability [70]. In these pathological alterations, Rg3 alleviated the reactive oxygen species (ROS) [71] and reduced the inflammatory cytokines, such as TNF-α, IL-1β, IL-6, and IL-13 [72]. These therapeutic activities of Rg3 have been proved to treat many diseases, such as liver diseases, neuroinflammatory diseases, and metabolic diseases [73,74,75]. These multifaceted properties of Rg3 may be involved in the inhibitory mechanisms of tumor metastasis.

To date, there was one set of clinical data showing a significant improvement in the 5-year metastatic-free rate more than twice (68.2% vs. 33.3% in the non-combination control) with the combination of *P. ginseng* and chemotherapy in gastric cancer patients [76]. Although, there is no Rg3-derived anti-metastatic clinical evidence yet, this clinical result may indicate the potential benefit of *P. ginseng*, including Rg3 on survival and anti-metastasis, especially as a format of combination therapy with existing agents. The present experimental data would support the potential of Rg3 decreasing the tumor metastatic risk.

The present study has some limitations. First, there were no clinical data on anti-metastatic effects of Rg3 supporting our results. The number of recently published studies is very small, and our study restricted the language to English only. Second, the doses of Rg3 from in vitro studies used were relatively high (up to 200 µg/mL), which might have a limited reliability. We, however, notice that four out of 10 in vitro studies are the supporting data of animal studies and the molecular weight of Rg3 (784.3 kDa) [77] is six times larger than 5-fluorouracil (5-FU) (130.1 kDa) [78]. Third, only one study engaged the positive control [37], so it is difficult to compare the anti-metastatic effect of Rg3 with other existing agents.

In spite of the above limitations, our results are valuable reference data for Rg3-derived studies targeting tumor metastasis. Rg3 has the advantage of safety that the no observed adverse effect level (NOAEL) is 180 mg/kg, and the lethal dose 50 percent (LD50) is above 800 mg/kg in Sprague Dawley (SD) rats [79]. Two clinical studies also reported on the safety that oral administration of Rg3 up to 50 mg/day causes no toxicity on patients with non-small cell lung carcinoma [80] and hepatocellular carcinoma [81]. Regarding pharmacokinetics of Rg3, the half-life of Rg3 was 14 min [82] and 18.5 min [83], both in SD rats after intravenous administration. On the other hand, Rg3 has been known to have a very low oral bioavailability (2.63%) [83,84] compared to other ginsenosides, including Rh2 (24.8%) [85], and Rg1 (18.4%) in SD rats [86]. This might have resulted from Rg3 being rapidly metabolized in the gastrointestinal tract through hydrolysis in the stomach [87]. These characteristics of Rg3 should be considered in future Rg3-derived drug development.

Taken together, we carefully suggest that Rg3 may provide an opportunity to develop novel anti-metastatic agents through further well-designed investigations and clinical studies.

## Figures and Tables

**Figure 1 ijms-23-09077-f001:**
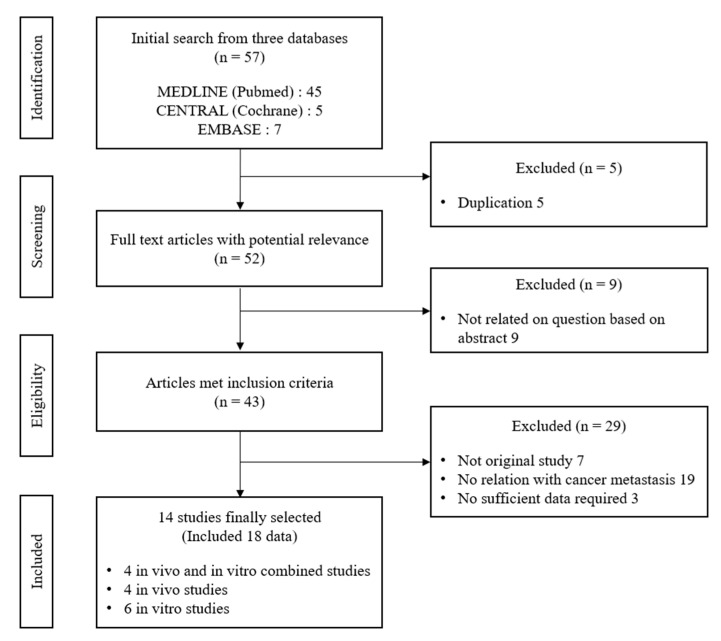
Flow diagram of literature search process.

**Figure 2 ijms-23-09077-f002:**
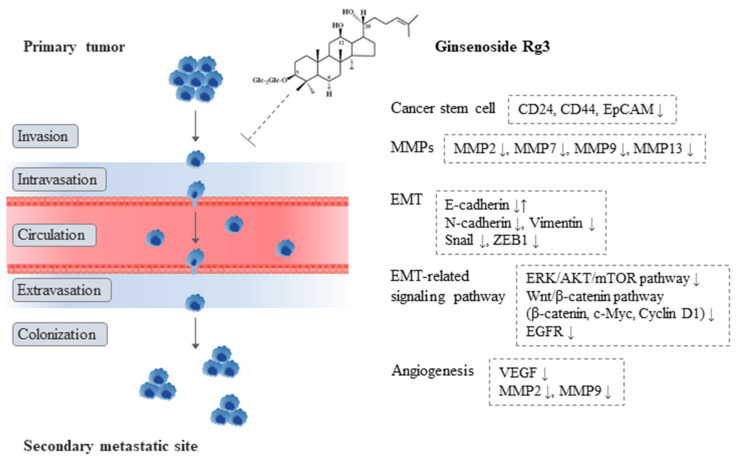
Anti-metastatic actions of ginsenoside Rg3.

**Table 1 ijms-23-09077-t001:** Characteristics of 14 included studies.

Items	In Vivo + In Vitro	In Vivo	In Vitro	Total
Number of study	4	4	6	14
Number of data	8	4	6	18
Mean Rg3 dose (days)	27.0 ± 26.1 mg/kg orally, 4.0 ± 1.0 mg/kg subcutaneously (28 ± 28 days)			
Rg3 type *				
Rg3	6	2	2	10
20(R)-Rg3	2	1	3	6
20(S)-Rg3	0	1	1	2
NpRg3	0	1	0	1
Cancer type (Cell line)				
Colorectal cancer	4	1 (AOM induction)	0	5
Melanoma	2	1	1	4
Liver cancer (HCC)	0	1 (DEN induction)	1	2
Lung cancer (NSCLC)	0	0	2	2
Thyroid cancer	2	0	0	2
Nasopharyngeal cancer	0	0	1	1
Osteosarcoma	0	0	1	1
Ovarian cancer	0	1	0	1
Metastatic model				
SC injection	2	3 (1 AOM, 1 DEN)		5
Tail vein injection	2	1		3
Target organ *				
Lung	3	3		6 ^*^
Liver	2	0		2
Kidney	1	0		1
Peritoneum	0	1		1
Animal				
C57BL/6 mouse	1	2		3
BALB/c mouse	2	0		2
Athymic nude mouse	0	1		1
NSG mouse	1	0		1
Wistar rat	0	1		1
Country (N. of studies)				
China				8
Korea				4
Japan				2

* Duplicate data from the same study are indicated. NpRg3, Nanoparticle conjugated Rg3; AOM, azoxymethane; HCC, Hepatocellular carcinoma; DEN, Dimethyl nitrosamine; NSCLC, Non-small cell lung cancer; SC, subcutaneous; NSG, NOD scid gamma mouse.

**Table 2 ijms-23-09077-t002:** Summary of 14 studies on anti-metastatic effects of ginsenoside Rg3.

Author (Year)	Cancer Type (Cell Line)	Animal (Sex)	Metastatic Model	Target Organ	Rg3	Concentration ^#^/ Administration/Duration	Main Outcome	Mechanism of Actions
In vivo								
Mochizuki et al. (1995) [32]	Melanoma (B16-BL6)	C57BL/6, F	Footpad SC inj.	Lung	20(R)-Rg3, 20(S)-Rg3	1.5 mg/kg, PO 14 days	N. of metastatic colonies *	Angiogenesis ↓
Iishi et al. (1997) [33]	Colorectal cancer	Wistar rat, M	SC inj. of AOM	Peritoneum	Rg3	5.0 mg/kg, SC 100 days	N. of peritoneal metastasis	Angiogenesis ↓
Xu et al. (2008) [34]	Ovarian cancer (SKOV-3)	Athymic nude mouse	Tail vain inj.	Lung	Rg3	0.3 mg/kg, SC 20 days	N. of metastatic colonies * N. of invaded cells *	Angiogenesis ↓ MMP9 ↓
Tang et al. (2018) [35]	Colorectal cancer (LoVo, SW620)	BALB/c, F	Flank SC inj.	Liver	Rg3	25.0 mg/kg, PO 12 days	N. of metastatic nodules *	Cancer stem cell markers (CD24, CD44, EpCAM) ↓ Angiogenesis ↓
Wu et al. (2018) [36]	Thyroid cancer (C643)	BALB/c, F	Tail vain inj.	Lung	Rg3	10.0 mg/kg, PO 18 days	N. of metastatic nodules *	Angiogenesis ↓
Meng et al. (2019) [37]	Melanoma (B16)	C57BL/6, M	Footpad SC inj.	Lung	Rg3	0.3 mg/kg, SC 28 days	N. of metastatic nodules *	Angiogenesis ↓ MMP2 ↓, MMP9 ↓ ERK ↓, AKT ↓, mTOR ↓
Phi et al. (2019) [38]	Colorectal cancer (HT29)	NSG mouse	Tail vain inj.	Liver, lung, kidney	20(R)-Rg3	5.0 mg/kg, SC 28 days	N. of metastatic nodules *	-
Ren et al. (2020) [39]	HCC	C57BL/6, M	SC inj. of DEN	Lung	NpRg3	70.0 mg/kg, PO 5 days	N. of metastatic nodules *	-
In vitro								
Kim et al. (2014) [40]	Lung cancer (A549)				20(R)-Rg3	25.0 μg/mL	Wound-healing migration ability * N. of migrated/invaded cells *	E-cadherin ↓, Snail ↓ MMP2 ↓
Lee et al. (2015) [41]	Melanoma (B16F10)				Rg3	25.0 μg/mL	Wound-healing migration ability * N. of migrated/invaded cells *	MMP13 ↓
Tian et al. (2016) [42]	Lung cancer (A549, H1299, H358)				20(R)-Rg3	25.0 μg/mL	Wound-healing migration ability * N. of migrated/invaded cells *	MMP2 ↓, MMP9 ↓ E-cadherin ↑, Snail ↓, N-cadherin ↓, Vimentin ↓
Tang et al. (2018) [35]	Colorectal cancer (LoVo, SW620, HCT116)				Rg3	200.0 μg/mL	Wound-healing migration ability * N. of migrated/invaded cells *	Cancer stem cell markers (CD24, CD44, EpCAM) ↓
Wu et al. (2018) [36]	Thyroid cancer (TPC1, BCPAP, C43, Ocut-2c)				Rg3	50.0 μg/mL	Wound-healing migration ability *	MMP2 ↓, MMP9 ↓ VEGF-A ↓, VEGF-C ↓
Meng et al. (2019) [37]	Melanoma (B16F1)				Rg3	5.0 µg/mL	Wound-healing migration ability * N. of migrated/invaded cells *	VEGF ↓ ERK ↓, AKT ↓, mTOR ↓
Wang et al. (2019) [43]	Nasopharyngeal cancer (HNE1, CNE2)				20(S)-Rg3	25.0 µg/mL	N. of migrated/invaded cells *	MMP2 ↓, MMP9 ↓ E-cadherin ↑, N-cadherin ↓, Vimentin ↓, ZEB1 ↓
Sun et al. (2019) [44]	Liver cancer (HepG2, MHCC-97L)				20(R)-Rg3	2.5 μg/mL	Wound-healing migration ability * N. of migrated/invaded cells *	ARHGAP9 ↑
Phi et al. (2019) [38]	Colorectal cancer (HT29, SW620)				20(R)-Rg3	10.0 μg/mL	N. of migrated/invaded cells *	MMP2 ↓ E-cadherin ↓, Snail ↓ EGFR ↓, AKT ↓
Mao et al. (2020) [45]	Osteosarcoma (MG63, 143B)				Rg3	50.0 μg/mL	Wound-healing migration ability * N. of migrated/invaded cells *	MMP2 ↓, MMP7 ↓, MMP9 ↓ N-cadherin ↓, Vimentin ↓ Wnt/β-catenin signaling pathway (β-catenin, c-Myc, Cyclin D1) ↓

AOM, azoxymethane; MMP, Matrix metallopeptidase; FUT4, Fucosyltransferase 4; EpCAM, Epithelial Cell Adhesion Molecule; VEGF, Vascular endothelial growth factor; ERK, extracellular signal-regulated kinase; AKT, Protein kinase B; mTOR, mammalian target of rapamycin; EMT, Epithelial-mesenchymal transition; ZEB1, Zinc Finger E-Box Binding Homeobox 1; ARHGAP9, Rho GTPase Activating Protein 9; EGFR, Epidermal growth factor receptor; NSG, NOD scid gamma; HCC, Hepatocellular carcinoma; DEN, Dimethyl nitrosamine; NpRg3, Nanoparticle conjugated Rg3 ^#^ The concentration indicated the lowest concentration showing the positive pharmacological activity. * Statistically significant results are indicated.; ↓: indicate that the mechanism is inhibited by Rg3. ↑: indicate that the mechanism is increased by Rg3.

## Data Availability

The data used for this study are available from the corresponding author upon request.

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
