# Peer review of "Experimental Evidence for the Anti-Metastatic Action of Ginsenoside Rg3: A Systematic Review"

_ijms, 2022, doi:10.3390/ijms23169077_

Round 1
Reviewer 1 Report
Thank you for submitting your manuscript. This comprehensive systematic review clearly demonstrates the anti-metastatic action of Ginsenoside Rg3.
Hopefully further studies will be performed in near future to fully understand and appreciate the anti-metastatic action of Ginsenoside Rg3.
Author Response
We sincerely appreciate reviewer for the thorough review.
Reviewer 2 Report
In the manuscript “Experimental Evidence for the Anti-Metastatic Action of Ginsenoside Rg3: A Systematic Review”, Oh et al explore the potential value of Ginsenoside Rg3 on tumor metastasis and the mechanisms of anti-metastatic based on experimental evidence. The authors searched all the related research articles base on three research databases, and a total of 14 studies, including 8 animal and 6 in vitro, were identified and used in the further analysis. The authors found that the major anti-metastatic mechanisms of Rg3 involve cancer stemness, EMT behavior, and angiogenesis. This work provides valuable reference data for Rg3-derived studies targeting tumor metastasis. This topic is interesting, and this review has substantial content. There are several suggestions to help the authors improve their manuscript.
How many languages were checked in this analysis? Only focus on English? whether considered other main languages, such as Korean, Chinese, and Japanese?
The authors only checked the papers published in the public journal. Whether the authors considered other presentation forms of the research and clinical achievements, such as patent, and conference reports?
How about the potential side effect of ginsenoside Rg3? Features of pharmacokinetics and toxicology are the focus of which most readers and patients concerned.
Author Response
In the manuscript “Experimental Evidence for the Anti-Metastatic Action of Ginsenoside Rg3: A Systematic Review”, Oh et al explore the potential value of Ginsenoside Rg3 on tumor metastasis and the mechanisms of anti-metastatic based on experimental evidence. The authors searched all the related research articles based on three research databases, and a total of 14 studies, including 8 animal and 6 in vitro, were identified and used in the further analysis. The authors found that the major anti-metastatic mechanisms of Rg3 involve cancer stemness, EMT behavior, and angiogenesis. This work provides valuable reference data for Rg3-derived studies targeting tumor metastasis. This topic is interesting, and this review has substantial content. There are several suggestions to help the authors improve their manuscript.
How many languages were checked in this analysis? Only focus on English? whether considered other main languages, such as Korean, Chinese, and Japanese?
The authors only checked the papers published in the public journal. Whether the authors considered other presentation forms of the research and clinical achievements, such as patent, and conference reports?
=> We really appreciate reviewer for the professional review. Regarding issues language and un-academic publication reviewer mentioned, it would be better to include those literature resources, but we have used the public classical journals written in English for keep the data quality in the present study. This however would be a limitation of our study, thus we described it in this revised ‘Discussion’ section.
How about the potential side effect of ginsenoside Rg3? Features of pharmacokinetics and toxicology are the focus of which most readers and patients concerned.
=> As reviewer indicated, we added the features about pharmacokinetics and toxicology of ginsenoside Rg3 in ‘Discussion’ section of our revised manuscript as follows;
Briefly,
Three pharmacokinetic studies of Rg3 exist, from only Spraguee-Dawley (SD) rats-derived data (but not human).
- The average half-life of Rg3 was 18.5 min under condition of the intravenous administration into SD rats (Qian et al. 2005). However, Rg3 was not detected in rat plasma collected after oral administration at 100 mg/kg (Qian et al. 2005).
- The half-life of 14 min after the intravenous administration into SD rats (Cai et al. 2003).
- Rg3 has low oral bioavailability of 2.63% in SD rats (Xie et al. 2005).
Two safety studies for Rg3 (one animal study and one clinical study) were reported.
- No observed adverse effect level (NOAEL) of Rg3 is 180 mg/kg, and the lethal dose 50 percent (LD50) is above 800 mg/kg in Spraguee-Dawley rats (Li et al. 2020).
- Oral administration up to 50 mg/day of Rg3 causes no toxicity in clinical studies on non-small cell lung carcinoma (Li et al. 2016) and advanced hepatocellular carcinoma (Zhou et al. 2016).
References
Qian, T.; Cai, Z.; Wong, R. N.; Mak, N. K.; Jiang, Z. H. In vivo rat metabolism and pharmacokinetic studies of ginsenoside Rg3. J. Chromatogr. B 2005, 816, 223-232.
Cai, Z.; Qian, T.; Wong, R. N.; Jiang, Z. H. Liquid chromatography–electrospray ionization mass spectrometry for metabolism and pharmacokinetic studies of ginsenoside Rg3. Anal Chim Acta X 2003, 492, 283-293.
Xie, H. T.; Wang, G. J.; Sun, J. G.; Tucker, I.; Zhao, X. C.; Xie, Y. Y.; Wang, W. et al. High performance liquid chromatograph-ic–mass spectrometric determination of ginsenoside Rg3 and its metabolites in rat plasma using solid-phase extraction for pharmacokinetic studies. J. Chromatogr. B 2005, 818, 167-173.
Li, C.; Wang, Z.; Li, G.; Wang, Z.; Yang, J.; Li, Y.; Gao, Y. et al. Acute and repeated dose 26-week oral toxicity study of 20 (S)-ginsenoside Rg3 in Kunming mice and Sprague–Dawley rats. J. Ginseng. Res. 2020, 44, 222-228.
Li, Y.; Wang, Y.; Niu, K.; Chen, X.; Xia, L.; Lu, D.; Kong, R.; Chen, Z.; Duan, Y.; Sun, J. Clinical benefit from EGFR-TKI plus ginsenoside Rg3 in patients with advanced non-small cell lung cancer harboring EGFR active mutation. Oncotarget 2016, 7, 70535.
Zhou, B.; Yan, Z.; Liu, R.; Shi, P.; Qian, S.; Qu, X.; Zhu, L.; Zhang, W.; Wang, J. Prospective study of transcatheter arterial chemoembolization (TACE) with ginsenoside Rg3 versus TACE alone for the treatment of patients with advanced hepatocellular carcinoma. Radiology 2016, 280, 630–639.
Reviewer 3 Report
This is a review article on the possible anti-metastatic action of ginsenoside Rg3 based on 14 in vitro and animal studies. One reference, which was published in 2002, indicated a benefit of the combinatorial application of ginseng with chemotherapy in gastric cancer patients. However, no clinical data are available in the last 20 years, and it is questionable whether ginsenoside is considered beneficial for clinical use. The review is based on a small number of references without any comparison with existing treatments.
Major
· Limited scope and references: The reference articles for this review are limited (14 in total), and some articles are published many years ago in 1966, 1995, 1997, 2000, etc. It is questionable whether this compound requires any attention.
· in vitro doses: the doses are mostly in the range of 50 – 100 ug/ml, and some study uses 200 ug/ml. These doses are much higher than most of the other agents, indicating the efficacy is extremely low.
· inhibitors of MMPs and angiogenesis: Logically, the review is flawed. The current strategy for inhibiting MMPs and angiogenesis in existing treatments is not appreciated in this review. However, the beneficial effects of Rg3 on their inhibitions are described.
· Comparison with other agents: it is recommended to present the comparisons of ginsenoside with existing agents, and show the rationale why existing agents do not work but ginsenoside does.
Author Response
This is a review article on the possible anti-metastatic action of ginsenoside Rg3 based on 14 in vitro and animal studies. One reference, which was published in 2002, indicated a benefit of the combinatorial application of ginseng with chemotherapy in gastric cancer patients. However, no clinical data are available in the last 20 years, and it is questionable whether ginsenoside is considered beneficial for clinical use. The review is based on a small number of references without any comparison with existing treatments.
=> We sincerely appreciate reviewer for the thorough review. To date, a lot of preclinical and clinical studies of Rg3 on ‘anticancer’ activities have been published, however, only a few studies have been conducted on the 'anti-metastatic’ effect. We have tried to extract valuable information focusing on ‘anti-metastatic’ effect of Rg3 in present review, based on the clinical impact of ‘metastasis’ and potential of Rg3 on it.
Major
Limited scope and references: The reference articles for this review are limited (14 in total), and some articles are published many years ago in 1966, 1995, 1997, 2000, etc. It is questionable whether this compound requires any attention.
=> We fully understand the reviewer’ critique. We however aimed to contribute to study ‘anti-metastatic’ effect using Rg3 in present review, based on the clinical impact of ‘metastasis’ and potential of Rg3 on it. We added this limitation in ‘Discussion’ section of revised manuscript.
in vitro doses: the doses are mostly in the range of 50 – 100 ug/ml, and some study uses 200 ug/ml. These doses are much higher than most of the other agents, indicating the efficacy is extremely low.
=> We really thank reviewer for the professional comments. As reviewer indicated, in vitro studies adapted relatively high doses (50-200 ug/mL), which might have a limited reliability. Four of 10 in vitro studies are the supporting data of animal studies, and there would be a reason that the molecular weight of Rg3 (784.3) is 6-time larger than 5-FU (130.1). In addition, we choose the highest concentration having efficacy in table 2, then we have revised it as the lowest concentration showing the positive pharmacological activity.
inhibitors of MMPs and angiogenesis: Logically, the review is flawed. The current strategy for inhibiting MMPs and angiogenesis in existing treatments is not appreciated in this review. However, the beneficial effects of Rg3 on their inhibitions are described.
=> We fully understand the reviewer’s critique. The fact that MMPs and angiogenesis were involved in the process of cancer metastasis is well-known. As reviewer indicated, however, the clinical efficacy of MMPs and angiogenesis inhibitors are controversial as their clinical trials were failed. We have revised the parts for inhibition of MMPs and angiogenesis in metastasis as adding the limitation of them.
Comparison with other agents: it is recommended to present the comparisons of ginsenoside with existing agents, and show the rationale why existing agents do not work but ginsenoside does.
=> We completely understand the reviewer’ critique. Unfortunately, only one study engaged the positive agent (20 mg/kg 5-FU has similar antitumor effect to 3.0 mg/kg Rg3 on the B16 melanoma cell line, Meng et al. 2019), so it is difficult to compare the effect of Rg3 with the existing agents. We demonstrated this as another limitation in ‘Discussion’ section of revised manuscript.
Reference
Meng, L.; Ji, R.; Dong, X.; Xu, X.; Xin, Y.; Jiang, X. Antitumor activity of ginsenoside Rg3 in melanoma through downregulation of the ERK and Akt pathways. Int. J. Oncol. 2019, 54, 2069–2079.
Round 2
Reviewer 2 Report
I appreciate that the authors have addressed the comments. The revised version of the manuscript “Experimental Evidence for the Anti-Metastatic Action of Ginsenoside Rg3: A Systematic Review” has improved considerably.
Author Response
We sincerely appreciate reviewer for the valuable and positive review once again.
Reviewer 3 Report
Thank you for the responses to the original comments. The authors tried to justify the importance of this review article. However, the efficacy of ginsenoside for metastatic cancer is not sufficiently supported in the revised manuscript. The responses are overall insufficient and they do not change the weak tone of this manuscript. It could be of any significance in combination with the existing agents that was shown in 2002.
Author Response
Once again, we really appreciate reviewer for the professional review and helpful critique. Based on the reviewer’ suggestion, we have revised our discussion focusing on the potential of Rg3 by combination therapy with existing agents.
Round 3
Reviewer 3 Report
In the re-revised manuscript, the authors revised the discussion focusing on the potential of Rg3 by combination therapy with existing agents. The revised discussion does not strengthen current evidence, which is not sufficient to support the efficacy for metastatic cancer.